# Synergistic AML Cell Death Induction by Marine Cytotoxin (+)-1(*R*), 6(*S*), 1’(*R*), 6’(*S*), 11(*R*), 17(*S*)-Fistularin-3 and Bcl-2 Inhibitor Venetoclax

**DOI:** 10.3390/md16120518

**Published:** 2018-12-19

**Authors:** Cristina Florean, Kyung Rok Kim, Michael Schnekenburger, Hyun-Jung Kim, Céline Moriou, Cécile Debitus, Mario Dicato, Ali Al-Mourabit, Byung Woo Han, Marc Diederich

**Affiliations:** 1Laboratoire de Biologie Moléculaire et Cellulaire du Cancer, Hôpital Kirchberg, 9, rue Edward Steichen, L-2540 Luxembourg, Luxembourg; cristina.florean@lbmcc.lu (C.F.); michael.schnekenburger@lbmcc.lu (M.S.); dicato.mario@chl.lu (M.D.); 2Department of Pharmacy, Research Institute of Pharmaceutical Sciences, College of Pharmacy, Seoul National University, 1 Gwanak-ro, Gwanak-gu, Seoul 08826, Korea; krkim85@snu.ac.kr; 3College of Pharmacy, Chung-Ang University, 84 Heukseok-ro, Dongjak-gu, Seoul 06974, Korea; hyunjungkim@cau.ac.kr; 4Institut de Chimie des Substances Naturelles, CNRS UPR 2301, Univ. Paris-Sud, University of Paris-Saclay, 1, Avenue de la Terrasse, 91198 Gif-Sur-Yvette, France; Celine.Moriou@cnrs.fr (C.M.); ali.almourabit@cnrs.fr (A.A.-M.); 5LEMAR, IRD, UBO, CNRS, IFREMER, IUEM, 29280 Plouzané, France; cecile.debitus@ird.fr

**Keywords:** acute myeloid leukemia, ABT-199, Mcl-1, bromotyrosine, (+)-11(*R*), 17(*S*)-fistularin-3, configuration, anticancer drug combination

## Abstract

Treatment of acute myeloid leukemia (AML) patients is still hindered by resistance and relapse, resulting in an overall poor survival rate. Recently, combining specific B-cell lymphoma (Bcl)-2 inhibitors with compounds downregulating myeloid cell leukemia (Mcl)-1 has been proposed as a new effective strategy to eradicate resistant AML cells. We show here that 1(*R*), 6(*S*), 1’(*R*), 6’(*S*), 11(*R*), 17(*S*)-fistularin-3, a bromotyrosine compound of the fistularin family, isolated from the marine sponge *Suberea clavata*, synergizes with Bcl-2 inhibitor ABT-199 to efficiently kill Mcl-1/Bcl-2-positive AML cell lines, associated with Mcl-1 downregulation and endoplasmic reticulum stress induction. The absolute configuration of carbons 11 and 17 of the fistularin-3 stereoisomer was fully resolved in this study for the first time, showing that the fistularin we isolated from the marine sponge *Subarea clavata* is in fact the (+)-11(*R*), 17(*S*)-fistularin-3 stereoisomer keeping the known configuration 1(*R*), 6(*S*), 1’(*R*), and 6’(*S*) for the verongidoic acid part. Docking studies and in vitro assays confirm the potential of this family of molecules to inhibit DNA methyltransferase 1 activity.

## 1. Introduction

Acute myeloid leukemia (AML) is one of the main causes of death related to leukemia, with 70% of patients either not responding to treatment or eventually relapsing to secondary chemoresistant AML. Standard therapy has been almost unchanged for the last 40 years, and new drugs or drug combinations targeting the resistant fraction of patients are actively researched [1].

B-cell lymphoma (Bcl)-2 and myeloid cell leukemia (Mcl)-1 are two major anti-apoptotic proteins, which are frequently deregulated in cancer. The Bcl-2 family of proteins is the target of recently developed synthetic BH3 mimetics, namely ABT-263 (Navitoclax) and ABT-199 (Venetoclax) or natural compounds [2]. BH3 mimetics reverse the anti-apoptotic activity of Bcl-2 restoring sensitivity to apoptotic stimuli. By targeting specifically Bcl-2 activity and sparing Bcl-extra large (Bcl-xL), ABT-199 avoids the appearance of thrombocytopenia. Due to this more favorable pharmacological profile, ABT-199 was recently approved by the U.S. Food and Drug Administration (FDA) for the treatment of chronic lymphocytic leukemia [3] and small lymphocytic lymphoma patients (www.fda.gov), and is under evaluation for AML treatment [4]. However, resistance to this single agent has emerged, suggesting urgency for the development of combination treatments [5]. Recently, it has been shown by our group and others that ABT-199, in combination with drugs abrogating Mcl-1 expression, synergistically kills AML patient cells [6,7].

Marine organisms and sponges in particular constitute a precious source of biologically active compounds, which provide potential agents against cancer [8,9,10,11,12,13,14,15]. We previously reported marine compound heteronemin [16,17] and others [18,19,20,21] to show interesting anti-inflammatory and anticancer potential. Recently, we described the compound isofistularin-3 from *Aplysina aerophoba* to act as a cytostatic drug against a panel of cancer cell lines and sensitizes lymphoma cell lines to tumor necrosis factor-related apoptosis-inducing ligand (TRAIL) [22]. Fistularins are a family of sponge-derived brominated tyrosine found in various genera of sponges, comprising molecules with cytotoxic activity against cancer cells [23,24,25,26,27]. Recently, fistularin-3 was also recognized in cultures of the marine bacterium *Pseudovibrio denitrificans* Ab134 isolated from the sponge *Arenosclera brasiliensis* [28], which evokes a potential use of microorganisms for sustainable supplies of complex marine molecules in drug development.

Due to the similarities of ^1^H and ^13^C NMR spectra of fistularin stereoisomers [25] confusion exists regarding the absolute configurations of C11 and C17 carbons of fistularins. While the configuration of the verongidoic acid part was established as 1(*R*), 6(*S*) for C1 and C6, the absolute configuration at the two secondary carbinols C11 and C17 is rather complicated. This high molecular weight molecule presents four secondary alcohols that hampered the co-determination of their absolute configuration by chemical modification and NMR studies. Suitable crystals for X-ray studies have never been obtained for one of the stereoisomers of fistularin-3. Moreover, the carbon configuration also seems to vary depending on sponge species, and even mixtures can occur [25]. The name 11-epi-fistularin-3 has been used to identify a molecule isolated from *Aplysina oroides*, which was recognized as a 11*R* isomer of fistularin-3 extracted from *Aplysina cauliformis*, attributing to this latter the 11*S* configuration [25]. Finally, the C17 configuration of all fistularin-3 isomers remains still undefined and the term isofistularin-3 is mainly used to indicate a generic fistularin-3 isomer so far.

Here, we report the activity of the stereoisomer (+)-1(*R*), 6(*S*), 1’(*R*), 6’(*S*), 11(*R*), 17(*S*)-fistularin-3 (also named (+)-11(*R*), 17(*S*)-fistularin-3, and abbreviated here as RS-F3), isolated from the sponge *Suberea clavata*. We characterized the complete absolute configuration of RS-F3, revealing that this compound is in fact the (+)-11(*R*), 17(*S*)-fistularin-3 isomer. We confirmed the potential of this isomer to interfere with DNA methyltransferase (DNMT) 1 and assessed its anticancer potential on a panel of AML cell lines and show that it induces differential proliferation arrest and cytotoxicity. We also found that RS-F3 induces an early decrease of Mcl-1 protein levels in sensitive AML cell lines, which was independent of Noxa, but could be rescued by proteasome inhibition. Moreover, combination treatments with ABT-199 resulted in strong induction of apoptosis in AML cell lines resistant to Bcl-2 inhibition, which was not observed in healthy human progenitor/stem CD34+ cells or platelets.

## 2. Results

### 2.1. Extraction, Purification and Determination of RS-F3 Absolute Configuration

The lyophilized sponge *Suberea clavata* (200 g) was extracted at room temperature with methanol/dichloromethane (MeOH/DCM) (1/1) to give 57.8 g of dried extract. The crude extract was submitted to n-butanol/H_2_O partition to obtain a desalted butanol extract (30 g). A portion (5 g) of the later extract was submitted to filtration on Phenomenex Sepra C18 (50 μm, 65 A) silica gel and eluted with a gradient H_2_O/MeOH (100/0 to 0/100) and MeOH/DCM (100/0 to 0/100) to give 6 fractions (f1 to f6). The fraction f3 (1.3 g) was submitted to reversed phase (Sunfire C18) HPLC chromatography with H_2_O + 0.1% HCOOH/CH_3_CN + 0.1% HCOOH: 85/15 to 0/100 in 40 min and yield 13 subfractions (F1 to F13). The fraction F6 (500 mg, retention time 21.8 min) was pure (+)-11(*R*), 17(*S*)-fistularin-3 (Figure 1). ^1^H and ^13^C data of the compound (+)-11(*R*), 17(*S*)-fistularin-3 (Table 1 and spectra in Appendix A) are provided.

### 2.2. Differential Cytostatic and Cytotoxic Activities of RS-F3

Four AML cell lines were treated with increasing concentrations of RS-F3 for up to 72 h. Results of trypan blue-exclusion assays showed that RS-F3 treatment reduced proliferation and viability of U-937, HL-60 and THP-1 cells (Figure 2A). We then performed nuclear morphology analyses after 72 h of treatment, to investigate the modality of cell death induced by RS-F3 in these cell lines. Results showed the appearance of apoptotic nuclei in U-937, HL-60 and THP-1 cells (Figure 2B). Of note, proliferation and viability of HEL cells were not affected by any of the RS-F3 concentrations tested (Figure 2A,B). For each cell line, the concentrations reducing growth and viability to 50% were calculated and reported in Table 2.

Colony formation assays showed a decrease of total cell growth in RS-F3-treated U-937, HL-60, and THP-1 at 25 µM, whereas the growth of HEL cells was not affected, which is in line with the other data (Figure 2C). To assess differential toxicity, we used healthy peripheral blood mononuclear cells (PBMCs). 100 µM RS-F3 reduced cell viability by about 14% (Figure 2D, upper panel). These results were confirmed by a flow cytometry analysis of the distribution of the lymphocyte population from healthy donors showing a minor, although significant, effect of the treatments (Figure 2D, lower panel). We validated our results by RS-F3-treated healthy proliferating primary human CD34^+^ cells. After 44 h of treatment, the viability of CD34^+^ cells was only moderately affected by RS-F3, even at the highest concentrations (22.5% apoptotic cells at 100 µM) (Figure 2E).

### 2.3. RS-F3 Strongly Down-Regulates the Anti-Apoptotic Protein Mcl-1

To investigate involvement of molecular markers well-known to contribute to AML cell survival in the effect of RS-F3, we assessed the expression of the anti-apoptotic protein Mcl-1, associated with resistance to chemotherapy in AML [29,30]. Kinetic analysis revealed that Mcl-1 protein levels were decreased in U-937 cells from 2 h of treatment (Figure 3A). Since proteasome-mediated Mcl-1 degradation is a well-described mechanism of Mcl-1 regulation [31], we co-treated U-937 cells with proteasome inhibitors MG132 and bortezomib to rescue Mcl-1 levels. In the presence of proteasome inhibitors, Mcl-1 decrease was prevented by 30% and 60% suggesting that decreased Mcl-1 levels are in part due to proteasomal degradation (Figure 3B). At the same time, the protein levels of the Mcl-1 destabilizer Noxa remained unchanged (Figure 3A), suggesting that Noxa is not involved in the observed Mcl-1 downregulation. A similar decrease of Mcl-1 protein levels was also observed in two other AML cell lines, OCIAML-3, and HL-60 (Figure 3C). Interestingly, Mcl-1 protein levels remained unaffected in AML M6 erythroleukemia HEL cells, which were not affected by RS-F3 treatment (Figure 3D).

### 2.4. RS-F3 Triggers Endoplasmic Reticulum Stress in U-937 Cells

We previously showed that isofistularin-3 induced endoplasmic reticulum (ER) stress in RAJI Burkitt lymphoma cells [22] and ER stress-inducing agents were shown to lower Mcl-1 levels [32,33]. Accordingly, we analyzed the expression levels of proteins involved in the unfolded protein response (UPR) following ER stress, in U-937 cells. Kinetic analyses in cells treated with 15 µM RS-F3 revealed a rapid two-fold increase of the phosphorylated form of protein kinase RNA (PKR)-like ER kinase (PERK). Increased PERK phosphorylation was apparent after 2 h, maintained after 24 h of incubation and followed by a decrease of glucose related protein 78 (GRP78), starting from 4 h. Moreover, the stress sensor protein sestrin-2 was upregulated in a time-dependent manner, starting at 4 h of treatment (Figure 4A). We next performed PCR analysis of X-box binding protein 1 (XBP1) mRNA, which is spliced by the inositol-requiring enzyme-1 (IRE1) pathway of the UPR upon activation. Analyses revealed the presence of the spliced form of XBP1, starting from 16 h of treatment (Figure 4B). Finally, C/EBP homologous protein (CHOP) mRNA levels were increased after 24 h treatment, further confirming the ER stress-inducing capacity of RS-F3 (Figure 4C). Thapsigargin was used in all experiments as a positive control for ER stress induction.

### 2.5. RS-F3 Sensitizes Bcl-2-Expressing AML Cell Lines to ABT-199

Since combination treatments targeting Mcl-1 and Bcl-2 are considered a promising anticancer strategy against AML [6,7,34], we tested the effect of a combination of RS-F3 with the clinically approved Bcl-2 inhibitor ABT-199 (Venetoclax) in various AML cell lines. We used U-937 and OCIAML-3 cell lines, both expressing high levels of Mcl-1 and Bcl-2 proteins, and the Bcl-2-negative TF-1 cell line as a negative control (Figure 5A). All cell lines tested are resistant to ABT-199 treatment, with IC_50_ above 1 µM [6,35]. We pre-treated cells with 15 µM RS-F3 for 20 h, and then 100 nM ABT-199 was added for an additional 24 h of incubation. Results demonstrated that U-937 and OCIAML-3 cells underwent massive apoptosis upon combination treatment. On the other hand, the Bcl-2-negative TF-1 cells are not sensitized by the same combination (Figure 5B). Caspase activity assays in the presence or absence of the pan-caspase inhibitor zVAD-FMK, as well as Western Blot analyses, confirmed caspase 3 activation in U-937 cells upon combination treatment (Appendix A and Figure 5C).

Interestingly, in the presence of the caspase inhibitor, U-937 cells shifted to a caspase-independent apoptotic-like cell death, whereas apoptosis was fully prevented, and a small fraction of necrotic cells appeared in the case of OCIAML-3 cells (Appendix A). Under the same conditions, Mcl-1 protein level was decreased in U-937 cells, whereas Bcl-2 protein level remained unchanged, with the appearance of a caspase-cleaved band, which disappeared in the presence of zVAD-FMK (Appendix A and Figure 5C). Finally, we evaluated whether concomitant treatments could produce the same effect as the sequential one. Interestingly, data showed a synergistic appearance of apoptosis in U-937 after 18 h of treatment with the two drugs applied at the same time (Appendix A).

We also investigated the effect of the combination on a healthy cell model. When 15 µM RS-F3 and 100 nM ABT-199 were added to human primary CD34^+^ cells from healthy donors, a reduction of cell viability of about 30% was observed, which is less than a half of the cell death occurring in the Bcl-2-positive AML cell lines tested (Figure 5D) witnessing a differential toxicity compared to cancer cells. Finally, considering the high rate of thrombocytopenia observed in patients treated with ABT-263, we demonstrated that our new combination treatment failed to affect the viability of human platelets from healthy donors (Figure 5D).

### 2.6. RS-F3 Docking Studies and In Vitro Activity Assay on DNMT1

Our previous research demonstrated that the anticancer activity triggered by isofistularin-3 was initiated by an inhibition of DNMT1 activity leading to DNA demethylation [22]. Moreover, simulated docking models on DNMT1 predicted that isofistularin-3 binds to the DNA-interacting region of DNMT1 and in vitro assays revealed that it inhibits DNMT1 activity [22]. Since these effects might be implicated in the observed anticancer activity of isofistularin-3, we hypothesized that RS-F3, belonging to the same structural family, could share similar mechanisms of action. We thus explored the potential of RS-F3 to interact with DNMT1.

Four C11-C17 diastereoisomers for fistularin-3 can exist in theory, but not all the diastereoisomers have been discovered in Nature so far. To gain structural insights of these fistularin-3 stereoisomers on DNMT1 activity, we implemented simulated docking experiments on DNMT1 with four C11-C17 diastereoisomers with established 1(*R*) 6(*S*), 1’(*R*), and 6’(*S*) configuration (Figure 6A) and all the other 60 possible stereoisomers from six chiral centers of fistularin-3 (Appendix A), using the PatchDock server followed by the FireDock server [36,37]. Average global energy score of all 64 docking experiments was -36.04 and C11(*R*)-C17(*S*) exhibited lower global energy score than other stereoisomers (Appendix A). When we focused on four C11-C17 diastereoisomers of fistularin-3 with established 1(*R*) 6(*S*), 1’(*R*), 6’(*S*) configuration, RS-F3 showed the lowest global energy score (−44.34) when DNMT1 (PDB ID: 4WXX, residue 351-1600 construct) [38] was used as a target. RS-F3 was predicted to bind to the interface between the CXXC domain and the replication focus targeting sequence (RFTS) domain (Figure 6B).

To further support the predicted binding modes of RS-F3 on DNMT1, we used a different docking program, AutoDock 4.2 [39], which is widely used for computer-aided drug design. According to the docking results with flexible ligands and flexible residues obtained by AutoDock 4.2, RS-F3 was predicted to bind to the methyl donor binding site and to the interface between CXXC and RFTS domains with similar estimated free energy of binding (−11.23 and −11.04 kcal/mol, respectively). Given the fact that increasing *S*-adenosyl methionine (methyl donor) concentration did not affect the inhibition of isofistularin-3 against DNMT1 activity, the methyl donor binding site would not be an isofistularin-3 binding site [22]. Taken all together, the interface between CXXC and RFTS domains would be a potent binding site of RS-F3.

In the case of most favorable docking pose, RS-F3 penetrated the negatively charged cleft that is surrounded by CXXC and RFTS domains and consolidated the interaction between CXXC and RFTS domains. Furthermore, RS-F3 stretched to the DNA binding loop on CXXC domain, SKQ, that could influence to the interaction between CXXC domain and DNA [40]. Given the fact that the conformational changes of CXXC domain are critical for the DNA recognition and DNMT1 activity [41,42], the interruption of CXXC domain movement by RS-F3 would diminish the enzymatic activity of DNMT1 (Figure 6B).

We then assessed the ability of RS-F3 to inhibit DNMT1 activity in vitro. We used epigallocatechin-3-gallate (EGCG), a well-known in vitro DNMT1 inhibitor, as a positive control. Results showed that 50 µM RS-F3 reduced the activity of purified DNMT1 by about 25% (Figure 6C).

## 3. Discussion

Finding new effective treatments against AML is one major challenge of current anticancer research. Several marine compounds have already shown promising activity against AML cells [43,44] and cytarabine, a part of the gold standard treatment in AML therapy, is itself derived from a product isolated from a marine sponge. Among sponge-derived bromotyrosine compounds, bastadins showed cytostatic and cytotoxic activity against cancer cells [45]. Moreover, psammaplin A has been previously shown to inhibit DNMT1, the enzyme responsible of DNA methylation maintenance during replication [46], and to induce cancer cell death. Since alterations in DNMT1 activity are involved in tumor development and progression, drugs that interfere with this enzymes are actively researched.

We report here an interesting cytostatic and cytotoxic effect against AML cell lines of a brominated sponge-derived compound of the fistularin family, named (+)-11*R*, 17*S*-fistularin-3, briefly RS-F3. RS-F3 was tested for its anticancer effect on a panel of AML cell lines. Three AML cell lines decreased proliferation and viability at 5–15 µM. These results are similar to those obtained previously with the compound isofistularin-3 from *Aplysina aerophoba*, a stereoisomer of fistularin-3 with a non-specified configuration at C17 [22]. In this previous study we showed isofistularin-3 triggers cell cycle arrest in cancer cell lines, including AML cells. Moreover, we reported that the effects on cancer cell proliferation were accompanied by autophagy induction, as well as demethylation within the aryl hydrocarbon receptor gene promoter in the lymphoma RAJI cell line. Considering these findings, we further investigated molecular mechanisms of RS-F3, a molecule of the same structural family.

We thus investigated the anti-proliferative molecular mechanism of RS-F3 by assessing levels of Mcl-1. We showed that sub-toxic doses of RS-F3 reduced Mcl-1 levels in AML cells lines at early time points. However, in HEL cells, which continued to proliferate and survive to RS-F3 treatment, we detected no Mcl-1 decrease. The JAK2 V617 mutation status of this cell line could be implicated in this differential behavior; however, this hypothesis remains to be tested. Overall, these data suggest an important role for Mcl-1 in the anti-proliferative effect of RS-F3. Mcl-1 downregulation could result from transcriptional inhibition, increased degradation or from the combination of these two phenomena. When U-937 cells were treated with two different proteasome inhibitors together with RS-F3, Mcl-1 levels were rescued. Our data suggest a contribution of proteasomal degradation to RS-F3-induced Mcl-1 decrease.

Importantly, in this study we found that combination treatments of RS-F3 with the Bcl-2 inhibitor ABT-199 sensitized U-937 and OCIAML-3 cell lines, but not Bcl-2 negative TF-1 cells or healthy platelets (and to minor extent healthy CD34^+^ cells), to sub-micromolar concentrations of ABT-199. These data attest for a promising sensitizing activity of this marine compound towards Bcl-2 inhibitors, which could be exploited in the treatment of AML with high Mcl-1 expression and resistance to BH3 mimetics.

From the drug development point of view, we showed here that RS-F3 produces anticancer effects both alone and in combination with a clinically approved compound, when used at 15 µM. Even if a working concentration in the low micromolar or nanomolar range is usually considered more favorable, we report here that such concentrations of RS-F3 are not toxic against the healthy cell models tested. Moreover, the working concentration of this new drug is comparable to those commonly used for the well-known chemotherapeutic agent cisplatin in cellular studies [47,48,49]. Other approved anticancer drugs, such as carboplatin, cyclophosphamide, bleomycin, or the histone deacetylase (HDAC) inhibitor Belinostat, are used at maximum plasma concentrations in the high micromolar range [50]. Moreover, over the past few years, combination therapies gained increasing interest, especially in the oncology field as they usually reduce the development of drug resistance frequently observed with single treatments [51]. Most interestingly, BH-3 mimetics like ABT199 most favorably synergize with cell stress inducers including cardiac glycosides [6] or coumarin derivatives [52] thus indicating that compounds triggering ER stress strongly precondition cancer cells towards pro-apoptotic inducers affecting the intrinsic, mitochondrial cell death pathway. Our previously published data already showed an interesting synergistic effect of isofistularin-3 with TNF-related apoptosis-inducing ligand (TRAIL), an activator of the extrinsic cell death pathway [22] so that we believe the fistularin family of compounds deserves to be further explored for its synergistic anti-cancer-induction, independently of the two main apoptotic pathways.

By our docking and in vitro studies, we identified DNMT1 as a target of RS-F3. This enzyme is also the target of the DNA demethylating agent Decitabine, which is a clinically approved treatment for AML patients. Drugs inhibiting DNMT1 lead to reactivation of tumor suppressor genes, thus restoring the cell sensitivity to endogenous death-inducing stimuli. Nowadays, only a few compounds have been reported as DNMT1 inhibitors, and new drugs are actively searched. For this reason, we consider RS-F3 as an interesting molecule, which could pave the way to further development of DNMT1 targeting compounds. We have to take into account that the high molecular weight of RS-F3 could constitute a challenge for the translation into the clinic; however, well-known anticancer drugs such as vincristine or paclitaxel, with molecular weight above 800 Daltons, were successful in meeting this challenge. Thus, RS-F3 constitutes an interesting scaffold for structure-activity relationship studies, as well as a potential sensitizing drug in combination treatments for resistant AML.

The hypothesis that RS-F3 could induce ER stress and elicit an UPR was also explored. Our previous data showed that ER stress is part of the mechanism of isofistularin-3-induced TRAIL sensitization in RAJI cell. We demonstrate here that RS-F3 induces also ER stress in U-937, by showing the activation of the PERK and IRE1 UPR pathways, as well as a decrease of GRP78 protein expression levels. This latter result is similar to what was obtained previously in the same cell line with the kinase inhibitor and ER stress inducer sorafenib [53]. PERK activation usually leads to reduced transcription in order to lower ER protein load and re-establish homeostatic conditions. Thus, globally reduced transcription levels could attest for a part of the Mcl-1 decrease seen upon RS-F3 treatments. The contribution of this pathway will be further investigated in future work.

It is known that UPR can alter the sensitivity of cancer cells to therapy, either by enhancing or by decreasing cell death, depending on the drug used and the cellular context [54]. Thapsigargin was shown to enhance taxane-induced cell killing in prostate cancer cells [55] and ritonavir, a human immunodeficiency virus protease inhibitor, induces ER stress and sensitizes sarcoma cells to bortezomib treatment [56]. However, UPR was also shown to promote topoisomerase II inhibitor resistance [57]. It would thus be important to determine the precise contexts and activation levels at which cellular stress can be exploited as a useful sensitizing strategy. Recently, the compound ONC201 was shown to induce an atypical integrated stress response, and to sensitize AML cells to ABT-199 [58]. In line with these results, our findings suggest a possible use of molecules targeting the cellular stress response systems to overcome ABT-199 resistance.

Bromotyrosine derivatives comprise several biologically active molecules [23,24,27,59]. We previously showed that isofistularin-3 docked into the DNMT1 DNA-binding CXXC domain and inhibits in vitro DNMT1 activity [22]. Docking was performed there using the crystal structures of DNMT1 in complex with the natural inhibitor sinefungin (PDB: 3SWR) or with DNA (PDB: 3PTA) [41], respectively. Interestingly, using the crystal structure of DNMT1 that includes an additional RFTS domain (PDB: 4WXX), RS-F3 was predicted to be the best diastereoisomer among 1(*R*), 6(*S*), 1’(*R*), 6’(*S*) fistularin-3 stereoisomers and to bind to the interface between CXXC and RFTS domains. In addition, RS-F3 exhibited an inhibitory effect on the in vitro DNMT1 activity. Altogether, these data support the increasing knowledge about the potential of marine bromotyrosines to modulate epigenetic enzymes and lead cancer cells to proliferation arrest or to death.

## 4. Materials and Methods

### 4.1. Compound Isolation and Structure Elucidation

#### 4.1.1. Sponge Collection and Compound Isolation

Compound RS-F3 was extracted from the freeze-dried sponge *Suberea clavata* (*Pulitzer-Finali, 1982*), which was collected by hand using SCUBA diving in Solomon Islands, Russel Group, off Lologhan Island on the 30/06/2004 between 6 m and 12 m deep. A voucher sample is deposited at the Queensland Museum under the accessing number G322641 and was identified by Dr. J.N.A. Hooper as *Suberea clavata* (*Pulitzer-Finali, 1982*). The sponge was deep frozen on board until work up.

The lyophilized sponge was extracted at room temperature with methanol/dichloromethane (MeOH/DCM) (1/1). The crude extract was submitted to n-butanol/H_2_O partition to obtain a desalted butanol extract. A portion of the extract was submitted to filtration on Phenomenex Sepra C18 silica gel and eluted with a gradient H_2_O/MeOH (100/0 to 0/100) and MeOH/DCM (100/0 to 0/100) to give 6 fractions. The fraction f3 was submitted to reversed phase (Sunfire C18) HPLC chromatography with H_2_O + 0.1% HCOOH/CH_3_CN + 0.1% HCOOH: 85/15 to 0/100 in 40 min and yield 13 subfractions (F1 to F13). The fraction F6 was pure (+)-11(*R*), 17(*S*)-fistularin-3.

#### 4.1.2. Determination of (+)-11(*R*), 17(*S*)-fistularin-3 Absolute Configuration

The absolute configuration of (+)-11(*R*), 17(*S*)-fistularin-3 was established using the Mosher method and intensive purification by RP-HPLC, LCMS and NMR analysis (Table 1 and Appendix A). The stereogenic carbinol centers C11 and C17 have been determined for the tetracylated Mosher esters C_71_H_58_^79^Br_3_^81^Br_3_F_12_N_4_O_19_Na, m/z 2000.84).

#### 4.1.3. Docking Studies

Docking simulations were processed using the PatchDock server [37] followed by the FireDock server [36] and AutoDock 4.2 program [39]. Docking template of DNMT1 were obtained from the Protein Data Bank (PDB ID: 4WXX), and the coordinates of 64 fistularin-3 stereoisomers were generated using the ChemDraw program (PerkinElmer Informatics, 16.0.1.4 version) in which stabilized forms were calculated with energy minimization to figure out stable conformation of fistularin-3 stereoisomers in solution. The energy minimization process was performed using Chem3D program with 0.01 minimum RMS gradient option. DNMT1 templates and coordinates of 64 fistularin-3 stereoisomers were used as receptor and ligand, respectively, in the PatchDock webserver operation. In the PatchDock options, clustering RMSD was configured as 4.0 and complex type was set as Enzyme-inhibitor. After coarse prediction of docking positions and scores according to the geometric shape complementarity in PatchDock, top 20 docking candidates were transferred to the FireDock operation for further rigid-body refinement of docking solutions. AutoDock 4.2 program was additionally used to predict binding modes of RS-F3 on DNMT1 with enhanced docking options, such as ligand and side-chain flexibility that is considered essential in recent computer-aided drug design [60]. After DNMT1 templates were modified by AutoDockTools [39] to include polar hydrogens and partial charges by computing the Gasteiger method, rigid body docking experiments of RS-F3 were initially implemented on candidate binding sites on DNMT1 by 126 × 126 × 126 points grid box with 0.3 angstrom spacing. Based on the initial docking results, we implemented refined docking experiments with flexible residues option for 5–6 residues that seemed to contribute to ligand binding, and a focused grid box was set by 60 × 60 × 60 points with 0.3 angstrom spacing. AutoDock 4.2 was performed with computational parameters as follows: Genetic Algorithm for search parameters, defaults for docking parameters, and Lamarckian GA for output option. Structural alignment and docking figures were produced by PyMOL [61].

### 4.2. Biological Assays

#### 4.2.1. Chemicals

EGCG, MG132, thapsigargin and VP16 were purchased from Sigma-Aldrich (Bornem, Belgium); bortezomib, Venetoclax (ABT-199), and ABT-263 were purchased from Selleckchem (Bio-Connect B.V., Huissen, The Netherlands). All drugs were dissolved in DMSO.

#### 4.2.2. Cell Proliferation and Viability Assays

The AML cell lines U-937, HL-60, HEL, THP-1, and TF-1 (Deutsche Sammlung von Mikroorganismen und Zellkulturen GmbH) were cultured in RPMI 1640 medium (Lonza, Verviers, Belgium) supplemented with 10% heat-inactivated fetal calf serum (FCS; Sigma-Aldrich) and 1% antibiotic–antimycotic (Lonza); TF-1 cells were also supplemented with 5 ng/mL granulocyte–macrophage colony-stimulating factor; PeproTech, London, UK). The AML cell line OCIAML-3 was cultured in MEM-ALPHA (Lonza) supplemented with 15% FCS and 1% antibiotic–antimycotic. PBMCs and platelets (Red Cross, Luxembourg, Luxembourg) from human healthy donors were isolated and cultured as previously described [6,62]. Human primary CD34^+^ progenitor/stem cells were isolated from cord blood of healthy donors and cultured as previously described [63]. All cells were cultured at 37 °C in humid atmosphere and 5% CO_2_.

Cell number and viability were determined using a semi-automated image-based Cedex cell counter (Innovatis AG, Roche, Basel, Switzerland) hinged on the Trypan Blue (Lonza) exclusion assay. Colony formation assays were performed as previously described [22], with a seeding concentration of 1000 cells/mL for U-937, THP-1, and HEL cells and of 500 cells/mL for HL-60 cells.

Lymphocyte subpopulation from PBMCs was identified by flow cytometry (FACSCalibur, BD Biosciences, Erembodegem, Belgium) based on cell size and granularity. Data were recorded statistically (10.000 events/sample) using the CellQuest 4.0.2 software (BD Biosciences, Erembodegem, Belgium) and analyzed using Flow-Jo 8.8.7 software (Tree Star, Inc., Ashland, OR, USA).

Platelet viability was assessed by measuring ATP levels with the Cell Titer-Glo Luminescent Assay (Promega, Leiden, The Netherlands), according to the manufacturer’s instructions. Data were recorded using an Orion Microplate Luminometer (Berthold Pforzheim, Germany). In parallel, viability was evaluated by measuring phosphatidylserine exposure by FACS analysis of cells stained with FITC Annexin V (FITC Annexin V Apoptosis detection kit, BD Pharmingen, Erembodegem, Belgium). The platelet-toxic compound ABT-263 and a challenge with 50% DMSO for 1 min were used as a positive control for platelet mortality.

#### 4.2.3. Evaluation of Cell Death Type and Caspase-3/7 Activity Assay

The modality of cell death was assessed performing nuclear morphology analyses by fluorescence microscopy (Olympus, Aartselaar, Belgium), after staining with Hoechst 33342 and propidium iodide.

Enzymatic activity of caspases-3/7 was determined using the Caspase-Glo 3/7 Assay (Promega, Leiden, The Netherlands). The assay was performed according to the manufacturer’s instructions and luminescence was measured using an Orion Microplate Luminometer (Berthold, Pforzheim, Germany).

#### 4.2.4. Protein Extraction and Western Blotting

Whole cell extracts were prepared using M-PER^®^ (Thermofisher, Erembodegen, Belgium) supplemented by 1× protease inhibitor cocktail (Complete EDTA-free, Roche, Basel, Switzerland) according to manufacturer’s instructions. Western blots were carried out, as previously described [64], using the following primary antibodies: Anti-Mcl-1 (4572S) and anti-PERK (3192) from Cell signaling (Leiden, The Netherlands); anti-sestrin-2 (10795-AP) from Proteintech (Sanbio, The Netherlands); anti-caspase 3 (sc-56053) and anti-GRP78 (sc-13968) from Santa Cruz Biotechnology (Boechout, Belgium); anti-Noxa (IMG349A) from Imgenex (Novus Biologicals, Cambridge, UK); anti-Bcl-2 (OP60) and anti-α tubulin (CP06) from Millipore (Merck, Brussels, Belgium); and anti-β actin (5441) from Sigma-Aldrich (Bornem, Belgium).

#### 4.2.5. Gene Expression and XBP1 Splicing Analyses

CHOP gene expression levels were evaluated by Q-RT-PCR, as previously described [65]. XBP1 splicing analysis was performed by end-point PCR. First, mRNA was extracted and retro-transcribed, as previously described [65]; then PCR was performed using HotStarTaq DNA Polymerase (Qiagen, Venlo, The Netherlands) following manufacturer’s instructions with 35 cycles of amplification and an annealing temperature of 60 °C. DNA amplicons were then resolved into a 15% polyacrylamide gel. Primer sequences are available on request.

#### 4.2.6. In Vitro DNMT1 Activity Assay

DNMT1 activity was measured using the in vitro DNMT activity/inhibition assay (Active Motif, Rixensart, Belgium) according to manufacturer’s instructions. The methylation reaction was performed by incubating 25 ng of purified DNMT1 with compounds for 2 h at 37 °C in the presence of 0.01% Triton X-100. The methylated DNA was then recognized by the His-tagged methyl-CpG binding domain protein 2b. The addition of a poly-histidine antibody conjugated to horseradish peroxidase then provided a colorimetric readout quantified with a spectrophotometer (SpectraCount, Packard, Groningen, The Netherlands) at the wavelength of 450 nm.

#### 4.2.7. Statistical Analysis

Statistical analyses were carried out using the GraphPad Prism 7.0 software (La Jolla, CA, USA). One-way or two-ways ANOVA, followed by the Holm-Sidak multiple comparison tests were used for statistical comparisons. Statistical significances were evaluated at *p*-values below 0.05. All histograms represent the mean ± SD of at least 3 independent experiments.

## 5. Conclusions

We determined the absolute configuration of the fistularin-3 stereoisomer (+)-11(*R*), 17(*S*)-fistularin-3 (RS-F3). We showed RS-F3 to induce an interesting anticancer effect against AML cell lines. Our data suggests an important role for Mcl-1 in mediating the anti-proliferative activity of the drug. Importantly, we show that this fistularin-3 stereoisomer synergizes with the Bcl-2 inhibitor ABT-199 in killing Bcl-2-positive AML cell lines. Moreover, our docking and in vitro studies support the increasing knowledge about the potential of marine bromotyrosines to modulate DNMT1 activity.

## Figures and Tables

**Figure 1 marinedrugs-16-00518-f001:**
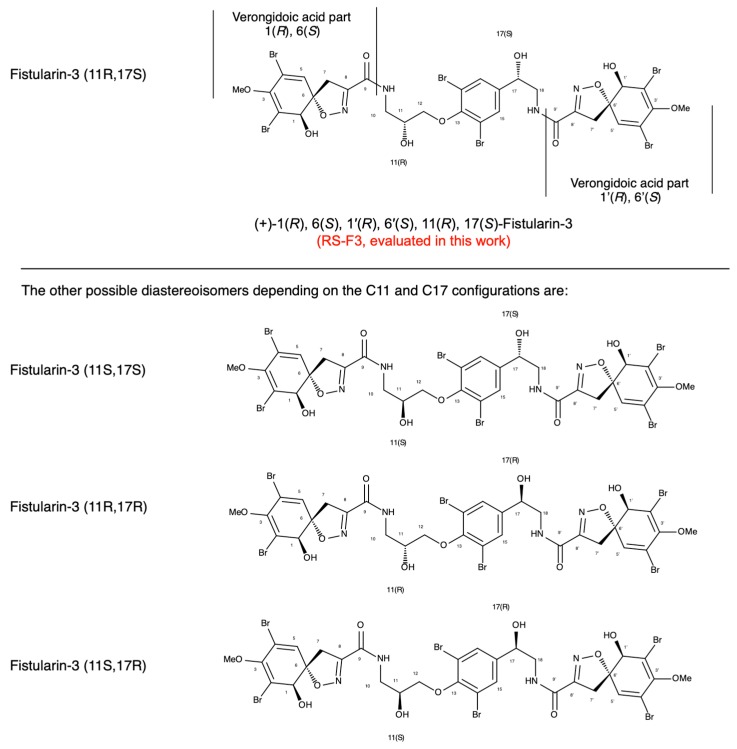
(+)-11R, 17S-fistularin-3: The configurations of C11 and C17 were determined by Mosher’s method. Preparation of *S*- and *R*-MTPA-11-epi-fistularin-3 esters, purification of the tetraacylated derivative by reversed phase-HPLC, NMR, and mass analyses.

**Figure 2 marinedrugs-16-00518-f002:**
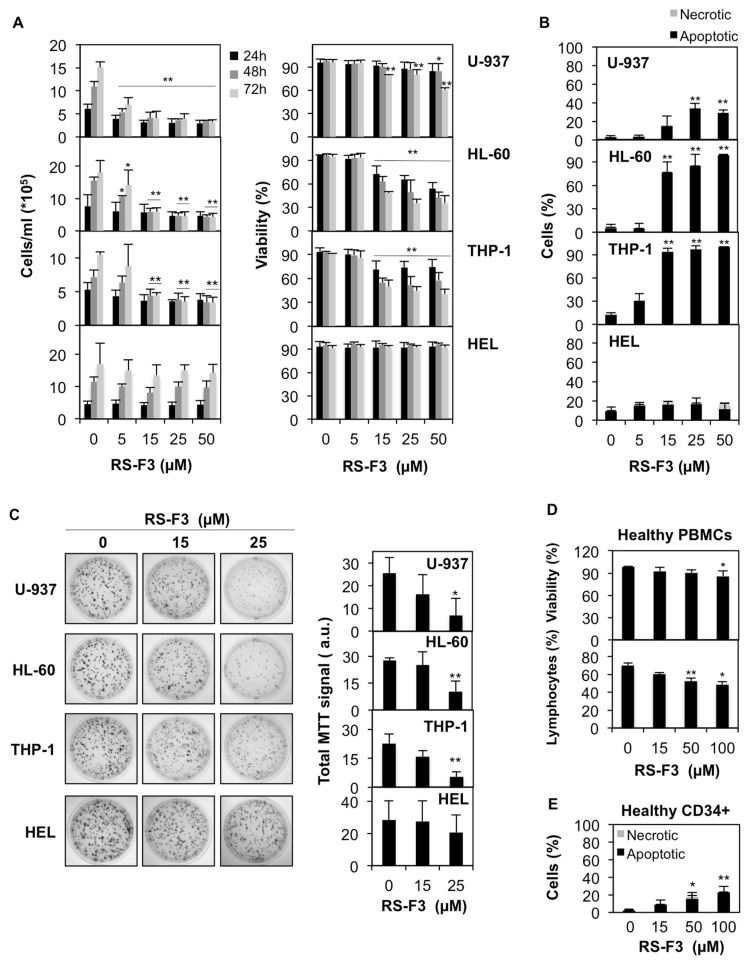
Anti-cancer effect of RS-F3 against AML cells. A panel of AML cell lines was treated with the indicated concentrations of RS-F3. (**A**) After 24, 48, and 72 h of treatment cell number and viability were determined. (**B**) After 72 h of treatment, the type of cell death was evaluated by nuclear morphology analyses. (**C**) Colony formation assays for the indicated cells lines treated with the indicated concentrations of RS-F3 (**left panel**) and the corresponding quantifications (**right panel**). Images are representative of three independent experiments. (**D**) Peripheral blood mononuclear cells (PBMCs) from healthy donors were treated with up to 100 µM RS-F3. After 44 h cell viability was analyzed by trypan blue staining (**upper panel**) or flow cytometry, based on forward scatter (cellular size) and side scatter (granularity) measuring the percentage of healthy lymphocytes (**lower panel**). (**E**) Healthy primary human CD34^+^ cells were treated with up to 100 µM RS-F3 for 44 h and nuclear morphology analyses were performed. All histograms represent the mean ± SD of at least 3 independent experiments. * *p* ≤ 0.05, ** *p* ≤ 0.01. a.u.: arbitrary units.

**Figure 3 marinedrugs-16-00518-f003:**
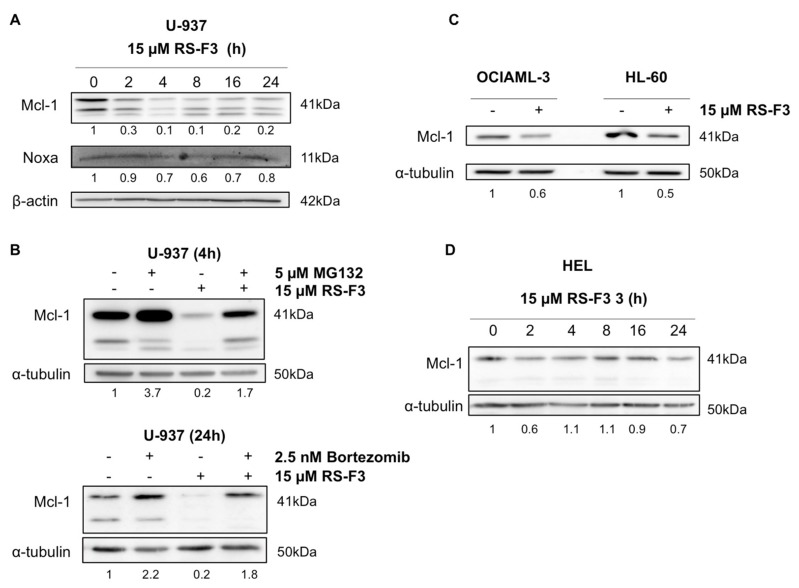
Effect of RS-F3 on Mcl-1 protein levels. (**A**) Western blot analyses in U-937 cells treated with 15 µM RS-F3 for the indicated time points. (**B**) Western blot analysis of Mcl-1 in U-937 cells treated for 4 h with or without 5 µM MG132, 15 µM RS-F3 or both drugs (**upper panel**), or for 24 h with or without 2.5 nM bortezomib, 15 µM RS-F3 or both drugs (**lower panel**). (**C**) Western blot analysis of Mcl-1 levels in OCIAML-3 and HL-60 cells treated for 4 h with 15 µM RS-F3. (**D**) Western blot analysis of Mcl-1 in HEL cells treated with 15 µM RS-F3 for the indicated time points. Blots are representative of three independent experiments.

**Figure 4 marinedrugs-16-00518-f004:**
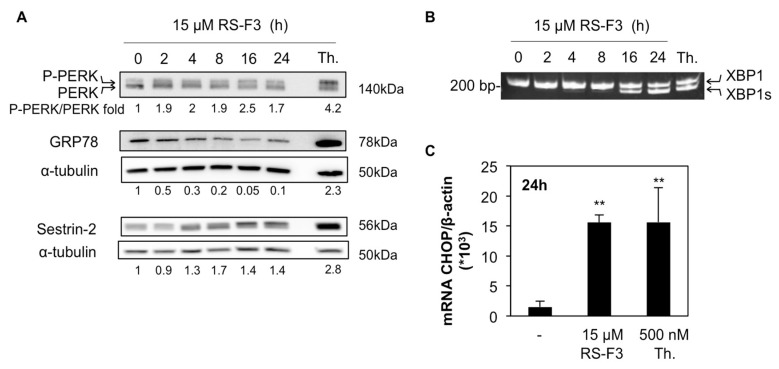
Effect of RS-F3 on ER stress markers in U-937 cells. U-937 cells were treated with 15 µM RS-F3 for the indicated time points followed by (**A**) Western blot analyses of PERK, GRP78, and sestrin-2; (**B**) end-point PCR analysis of XBP1 mRNA in its whole and spliced (XBP1s) forms; and (**C**) analyses of CHOP mRNA levels. Cells treated with 500 nM thapsigargin (Th.) were used as positive control for ER stress induction (2 h for XBP1 splicing, 24 h for other experiments). Histograms represent the mean ± SD of three independent experiments. Blots and gels are representative of three independent experiments. ** *p* ≤ 0.01.

**Figure 5 marinedrugs-16-00518-f005:**
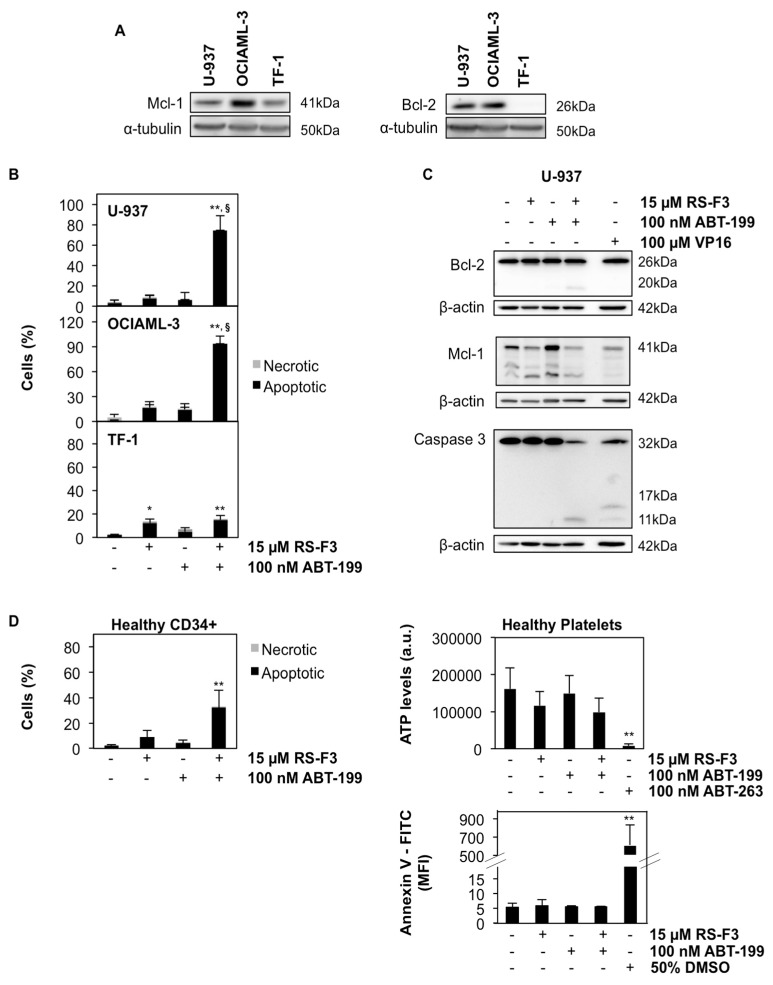
Effect of combination treatments of RS-F3 and ABT-199 in AML cell lines. (**A**) Western blot analysis showing basal levels of Mcl-1 and Bcl-2 in U-937, OCIAML-3, and TF-1 cells. (**B**) Cells were treated for 20 h with 15 µM RS-F3 followed by 24 h of 100 nM ABT-199, then nuclear morphology analysis was performed. (**C**) Western Blot analyses of Bcl-2, Mcl-1, and caspase 3 in U-937 treated as above. Cells treated with 100 µM VP16 for 3 h were used as a positive control for caspase 3 cleavage. (**D**) Nuclear morphology analysis of healthy CD34^+^ cells treated as above (**left panel**). CellTiter-Glo analysis (**right upper panel**) and annexin V staining (**right lower panel**) of healthy platelets treated as above. Cells treated with either 100 nM ABT-263 for 24 h or 50% DMSO for 1 min were used as positive control for platelet viability assays. Blots are representative of three independent experiments. Histograms represent the mean ± SD of at least three independent experiments. Asterisks indicate statistical difference with respect to control. The symbol § indicates statistical difference of combination treatments with respect to both compounds taken alone. * *p* ≤ 0.05, ** *p* ≤ 0.01; § ≤ 0.01. a.u.: arbitrary units, MFI: Mean fluorescence intensity.

**Figure 6 marinedrugs-16-00518-f006:**
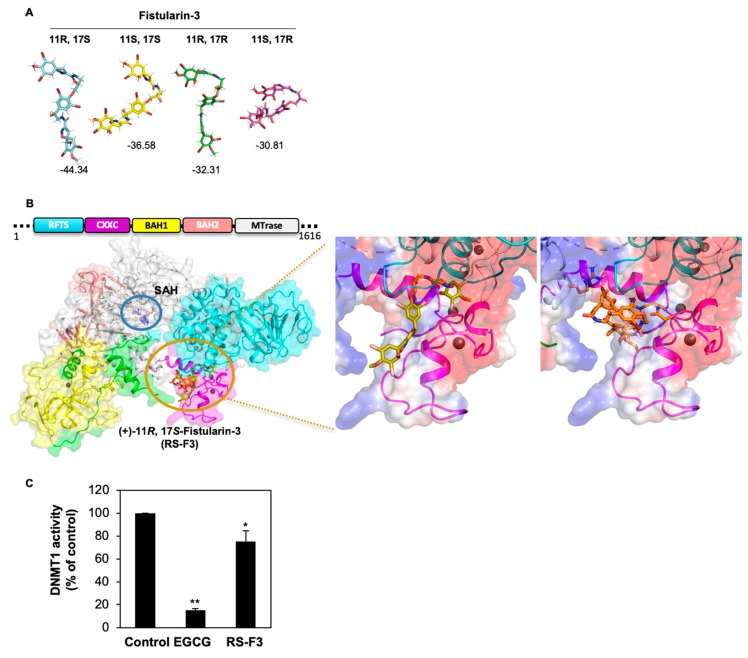
Docking of RS-F3 and diastereoisomers on DNMT1 and effect of RS-F3 on DNMT1 activity. (**A**) Energy-minimized structures of four diastereoisomers and predicted global energy scores of each diastereoisomer from PatchDock followed by FireDock. (**B**) Schematic diagram of DNMT1 domains and the best binding poses of RS-F3 on DNMT1 (PDB ID: 4WXX). The best RS-F3 binding poses from FireDock and AutoDock 4.2 are represented by stick models in gold and orange, respectively. RFTS domain, CXXC domain, BAH1 domain, BAH2 domain, methyltransferase domain, and linker are colored in cyan, magenta, yellow, pink, white, and green, respectively. The interface between CXXC and RFTS domains and the methyl donor binding site are circled in gold and blue, respectively (**left panel**). *S*-adenosyl homocysteine (SAH) bound to the methyl donor site is shown in blue stick model with label. The best binding poses of RS-F3 from FireDock (**middle panel**) and AutoDock 4.2 (**right panel**) are magnified on electrostatic potential surfaces of DNMT1. Negatively and positively charged surfaces of DNMT1 are represented by red and blue shade, respectively. (**C**) In vitro DNMT1 activity in the presence of 50 µM RS-F3 or the positive control epigallocatechin-3-gallate (EGCG). Data are reported as percentage of DNMT1 activity compared to the control. Histogram represents the mean ± SD of 3 independent experiments. * *p* ≤ 0.05, ** *p* ≤ 0.01.

**Table 1 marinedrugs-16-00518-t001:** ^1^H and ^13^C NMR data for (+)-11(*R*), 17(*S*)-fistularin-3 in Acetone-d6, 500 MHz, 298 K.

	Fistularin 3 (500 MHz)	
Position	δC, Type	δH Mult. (J in Hz)
1	75.2, CH	4.18, d (8.0)
2	122.1, C	
3	148.8, C	
4	113.8, C	
5	132.3, CH	6.52, s
6	91.8, C	
7	40.0, CH2	3.85, d (18.0)
		3.19, d (18.0)
8	155.1, C	
9	160.5, C	
10	43.6, CH2	3.80, m
		3.54, m
11	69.9, CH	4.25, m
12	75.9, CH2	4.06, dd (9.1, 5.5)
		4.02, dd (9.1, 5.5)
13	152.7, C	
14, 14’	118.4, C	
15, 15’	131.5, CH	7.66, s
16	143.3, C	
17	71.0, CH	4.90, ddd (7.7, 5.5, 4.3)
18	47.7, CH2	3.63, m
		3.49, m
1’	75.3, CH	4.19, d (8.0)
2’	122.1, C	
3’	148.8, C	
4’	113.9, C	
5’	132.4, CH	6.53, s
6’	91.8, C	
7’	40.0, CH2	3.82, d (18.0)
		3.16, d (18.0)
8’	155.2, C	
9’	160.5, C	
OMe	60.2, CH3	3.73, s
OMe’	60.2, CH3	3.73, s
NH		7.62, bt (6.0)
NH’		7.66, bt (6.0)
OH 1-1’		5.41, d (8.0)
OH 11		4.44, d (5.3)
OH 17		5.00, d (4.3)

**Table 2 marinedrugs-16-00518-t002:** Concentration of RS-F3 inhibiting 50% of the growth (GI_50_) or viability (IC_50_) of a panel of AML cell lines upon 72 h of treatment.

Cell Line	GI_50_ (µM)	IC_50_ (µM)
U-937	3.2	>50
HL-60	10.86	12.1
THP-1	15.17	10.36
HEL	>50	>50

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
