# Peer review of "Synergistic AML Cell Death Induction by Marine Cytotoxin (+)-1(R), 6(S), 1’(R), 6’(S), 11(R), 17(S)-Fistularin-3 and Bcl-2 Inhibitor Venetoclax"

_marinedrugs, 2018, doi:10.3390/md16120518_

Reviewer 1 Report

This manuscript describes the anticancer profile of one diastereomer of a marine cytotoxin when combined with a Bcl-2 inhibitor.  The biological evaluations are well done and presented. Weaknesses consisting of sensitivity issues associated with low sensitivity that is indicated by micromolar concentrations necessary to show activity, and the need to be in a combination with another anticancer agent to show activity, dampen a little the enthusiasm for the work. The confirmation of the absolute configuration of the isolated fistularin is useful information for further studies. The docking studies incrementally add information about this group of potential drug candidates. Mentioned is the possibility that other sponge species may produce other stereoisomers. Accordingly, docking studies to predict bioactivity of all stereoisomers could be useful. The description of AML treatment and fistularin development are well described, although the overall druggability of the fisularin molecule was not adequately discussed.  

Author Response

Please consider the attached Word document.

Reviewer 2 Report

Florean et. al., showed synergistic effect of fistularin-3 & ABT-199 on killing Mcl-1/Bcl-2-positive AML cell lines. They also resolved absolute configuration of carbons 11 & 17 of fistularin-3. Current study opens avenues for further research on synergistic treatments against leukemia. However, docking studies predict compound bind at 3 binding sites. Such a conclusion can be drawn by just reading about binding sites on DNMT1. The study has been performed with a minimal knowledge of the method and requires revision. I suggest referring to docking analysis papers like Recent advances in computer-aided drug design Curr Top Med Chem. 2014;14(16):1875-89 or similar.

Author Response

(The authors gave the same response as above.)

Reviewer 3 Report

Methods of isolation of fistularin-3 and its structure elucidation must be added briefly in the section of (method and material). The marine organism, better, to describe in subsection separated from (chemicals). Methods need to classify to into two main groups:

1.       isolation/structure elucidation

2.       biological assays and investigations

Explain the relationship between the chemical structure of fistularin-3 and its effects on AML cells.

Give list for all abbreviations

Conclusion section lacks.

Author Response

Please consider the attached Word document.

Round  2

Reviewer 1 Report

This manuscript has been successfully revised in response to reviewer comments.

Author Response

Please consider the attached document.

Reviewer 2 Report

Authors made a sincere effort to revise manuscript as per suggestions. I suggest the article for publication after the following changes

Note 1:

126*126*126 is a large grid box for docking and such results are not usually published. You almost performed blind docking rather than focused docking on a binding pocket of interest. 
If you had used something 60*60*60 that will given better results and the second best binding site might be the top ranked one in Autodock. This could have eliminated multiple binding site issue in the discussion.

Note 2:

Docking results based on 3SWR, which lacks one of the binding site may
not be worth presenting. Presenting 3 different binding sites also digresses binding site of interest. It may be worth combining two paragraphs (lines 239-273) into one. Results not consistent with experiments may be removed or placed into supplementary information. 

Note 3:
line 431 has two '.'
437 - 64 stereoisomers?
441-442 = to be rephrased

Change Gasteiger charge to Gasteiger method

Author Response

Please consider the attached document.

Reviewer 3 Report

The revised manuscript contains the corrected notes.

·         please, complete the biological name of the sponge: genus/ species (authority)/ family names (add in the method section)

Author Response

Please consider the attached document.
